# Gender Disparities of Heart Disease and the Association with Smoking and Drinking Behavior among Middle-Aged and Older Adults, a Cross-Sectional Study of Data from the US Health and Retirement Study and the China Health and Retirement Longitudinal Study

**DOI:** 10.3390/ijerph19042188

**Published:** 2022-02-15

**Authors:** Yifei Li, Yuanan Lu, Eric L. Hurwitz, Yanyan Wu

**Affiliations:** 1Thompson School of Social Work & Public Health, University of Hawai‘i at Mānoa, Honolulu, HI 96822, USA; liyifei@hawaii.edu (Y.L.); yuanan@hawaii.edu (Y.L.); ehurwitz@hawaii.edu (E.L.H.); 2Department of Disease Control and Prevention, Tang Du Hospital, Air Force Medical University, Xi’an 710038, China

**Keywords:** smoking, alcohol drinking, heart disease, gender, health survey

## Abstract

Heart disease remains the leading cause of death globally by gender and region. Smoking and alcohol drinking are known modifiable health behaviors of heart disease. Utilizing data from the US Health and Retirement Study and the China Health and Retirement Longitudinal Study, this study examines heart disease disparities and the association with smoking and drinking behavior among men and women in the US and China. Smoking and drinking behavior were combined to neither, smoke-only, drink-only, and both. In the US, the prevalence was higher in men (24.5%, 95% CI: 22.5–26.6%) than in women (20.6%, 95% CI: 19.3–22.1%) and a higher prevalence was found in the smoke-only group for both genders. In contrast, women in China had higher prevalence (22.9%, 95% CI: 21.7–24.1%) than men (16.1%, 95% CI: 15.1–17.2%), and the prevalence for women who smoked or engaged in both behaviors were ~1.5 times (95% CI: 1.3–1.8, *p* < 0.001) those who did not smoke or drink, but no statistical difference were found in men. The findings might be due to differences in smoking and drinking patterns and cultures by gender in the two countries and gender inequality among older adults in China. Culturally tailored health promotion strategies will help reduce the burden of heart disease.

## 1. Introduction

Heart disease remains the leading cause of death globally. The global evaluation of heart disease mortality showed that females had lower mortality rates of heart disease than males, and the differences declined with the increase of age [1]. The highest number of cardiovascular disease deaths occurred in China, followed by India, Russia, the US, and Indonesia [2]. It was estimated that the number of deaths due to cardiovascular disease was 17.9 million each year and is anticipated to rise to 20 million by 2030 worldwide [3]. In the US, the age-adjusted mortality of heart disease in 2017 was 209.0 among men and 129.6 among women per 100,000 population [4]; in China, the rates reported in 2016 were 101.42 for men and 76.55 for women, respectively [5].

The burden of heart disease in both countries is closely related to the increased mortality, morbidity, and frailty in the affected individuals, which could also translate to significant overall healthcare costs [6]. In addition, population aging has brought similar public health challenges of heart disease in the US and China [7]. Older adults experienced higher rates of chronic diseases and were particularly vulnerable as heart disease-related morbidity and mortality both increased with age [8,9]. According to the US National Center for Health Statistics, the prevalence of heart disease among men for different age groups were 9.5% (45–54 years), 16.7% (55–64 years), 29.8% (65–74 years), and 42.1% (75 years or older), while the prevalence among women were 9.4% (45–54 years), 13% (55–64 years), 19.3% (65–74 years), and 30.9% (75 years or older) [4]. However, no specific statistics of heart disease prevalence by age groups or gender were found for China. A previous study showed that the mortality rate of heart disease among men for different age groups was 94.9 (45–64 years old) and 875.4 (65+ years old) per 100,000 population per year, while the mortality rate among women was 41.8 (45–64 years old) and 866.8 (65+ years old) per 100,000 population per year in 2016 [5].

There are several risk factors for heart disease, which include socioeconomic status such as educational attainment; lifestyle behaviors such as alcohol use, cigarette smoking, unhealthy diet, restless sleep, and physical inactivity; other chronic diseases such as hypertension and diabetes; and body mass index (BMI) [10,11,12]. Several studies in multiple countries have shown that the mortality of coronary heart disease, the prevalence of cardiovascular major risk factors, and the health-related inequalities could be largely reduced by the change of unhealthy lifestyle behaviors [13,14,15]. Smoking and alcohol drinking are two commonly recognized unhealthy lifestyle behaviors in the US and China, and they are also among the top causes of preventable deaths [16]. Such behaviors not only pose health hazards to individuals but also increase the medical burdens to a greater extent. However, the management and control of these behaviors in the two countries are still below expectation. Therefore, the enhanced management of smoking and alcohol drinking behaviors could be an essential approach to reduce the unhealthy lifestyle-related chronic disease burden in our society [17]. Previous studies have shown that the associations between lifestyle behaviors and heart disease varied by age and gender. Coronary heart disease risk among current smokers was highest in the younger and lowest in the older participants in the US [18]. Gender differences existed in the effect-size measures of lifestyle-related factors, indicating that specific approaches were needed for men and women for enhanced prevention of primary and secondary cardiovascular disease management [19].

Population-based survey studies with large sample sizes will improve the current understanding of the roles of age and gender in the relationship between smoking and alcohol drinking behaviors and heart disease, and cross-country comparison will serve as validations of the findings and provide essential baselines for the development of health promotion and health care strategies for the global population [1]. Utilizing harmonized data from two sister surveys of the Gateway to Global Aging Data (https://g2aging.org/ (accessed on 1 July 2021)), the US Health and Retirement Study (HRS) and the China Health and Retirement Longitudinal Study (CHARLS), this study aimed to estimate the prevalence of heart disease among middle-aged and older adults (aged 50 years old and above) by gender and analyze their associations with smoking/alcohol drinking behaviors in the US and China. We hypothesize that smoking and alcohol drinking behaviors are associated with heart disease and the patterns of association differ by gender in both countries.

## 2. Materials and Methods

### 2.1. Data Source

The HRS is a biennial longitudinal study that utilized a complex multistage area probability sampling design of the US populations aged 50 and over and their spouses [20,21]. It has been conducted since 1992 and included a refresher cohort of 50–56 persons every 6 years. The HRS was sponsored by the National Institute of Aging (NIA U01AG009740) and is conducted by the University of Michigan. The data in HRS were collected by face-to-face interview or phone call, during which participants were asked questions about finances, health status and behaviors, marital/family status, and social support systems [21]. The CHARLS surveyed populations aged 45 years old and older and their spouses, including the assessments of social, economic, and health circumstances of community residents [22,23,24]. The baseline survey was conducted between June 2011 and March 2012, and participants were followed up every 2 years [25]. The study utilized the 2016 wave of HRS data from the RAND HRS 1992 to 2016 version 2 and the 2015 wave of Harmonized CHARLS version C and included those who were 50 years of age or older.

### 2.2. Analytical Sample

The original sample sizes were 20,912 in the 2016 wave of HRS data and 20,281 in the 2015 wave of CHARLS data. The exclusion criteria of this study were (1) participants aged less than 50 years old or missing age (*n* = 764 in HRS, *n* = 3860 in CHARLS); (2) missing heart disease status (*n* = 40 in HRS, *n* = 1560 in CHARLS); (3) missing of both smoking and alcohol drinking (*n* = 100 in HRS, *n* = 37 in CHARLS); and (4) observations with missing data on diabetes or high blood pressure (*n* = 76 in HRS, *n* = 495 in CHARLS). The sample sizes employed in this study were 19,932 in the HRS and 14,329 in CHARLS. The detailed sampling flow charts are shown in Figure 1.

### 2.3. Variables

The response variable heart disease (yes vs. no) was defined by the respondents’ self-reported answers to the question: “Has a doctor ever told you that you have had a heart attack, coronary heart disease, angina, congestive heart failure, or other heart problems?”.

The exposure variable was the combined smoking and alcohol drinking status, which was categorized as neither, smoke-only, drink-only, and both, based on the questions of whether the respondent reported ever smoked, or ever drank alcohol. The definition of smoking included more than 100 cigarettes in the lifetime in the US, and chewing tobacco, smoking a pipe, self-rolled cigarettes, cigarettes, or cigars in China. The drinking types referred to any kind of alcoholic beverages in the US and various alcoholic beverages, including white liquor and liang of liquor, in China.

Gender (men vs. women) was considered as a potential effect modifier because interaction between gender and smoking and alcohol drinking behaviors were previously found according to the literature reviews. Confounding variables included in the study were age group (50–59, 60–69, 70–79, and ≥80 years), education (less than high school, high school or equivalent (high school or vocational school), and the associate degree or higher (some college, associate degree, bachelor’s degree and above), household income tertiles, sleep quality (restless or good), high blood pressure and diabetes, BMI (normal: 18.5 ≤ BMI < 25 kg/m^2^, overweight: 25 ≤ BMI < 30 kg/m^2^, obese: ≥30 kg/m^2^, underweight: <18.5 kg/m^2^), and US race/ethnicity (White, Hispanic, Black, and other). Restless sleep was defined as how often they had trouble falling asleep most of the time in the HRS and ≥3 days a week in CHARLS. Similar to heart disease, high blood pressure and diabetes were self-reported doctor-diagnosed conditions. There was substantial missingness in CHARLS sleep quality, household income, and BMI data; therefore, we coded the missingness as a category “did not report” to ensure the population representativeness of the data.

### 2.4. Statistical Analysis

Statistical software R version 3.5.2 (Vienna, Austria) was used for the analyses. Separate analyses were performed for men and women in the US and China, respectively. We first computed descriptive statistics (frequency and weighted percentage) to summarize the sample characteristics. Next, we computed the weighted prevalence of heart disease by age group and smoking/drinking behavior, and the corresponding 95% confidence intervals (CIs) by country. Lastly, modified Poisson regression analyses were conducted to estimate the weighted crude prevalence ratio (cPR) and adjusted prevalence ratio (adjPR) of heart disease because odds ratios can substantially overestimate the prevalence ratios for common outcomes (when prevalence > 10%). The adjPRs adjusted for BMI, household income, educational level, sleep quality, high blood pressure, diabetes, BMI categories, and race/ethnicity. We also reported the 95% confidence intervals (CIs) for all PRs by age group (reference: 50–59 years) and drinking behavior (reference: neither smoking nor drinking), with the corresponding 95% CIs. The adjPRs were estimated by controlling for BMI, educational level, sleep quality, high blood pressure, and diabetes.

## 3. Results

### 3.1. Sample Characteristics

As shown in Table 1, there were 46.6% men in the US HRS sample and 48.7% men in the CHARLS sample. In the US, 40.7% of men and 31.3% of women reported yes to both smoking and drinking behaviors, and 12.8% of men and 24.3% of women reported no to neither behavior. As for China, 61.1% of men and only 2.9% of women reported both smoking and drinking, while 6.9% of men and 71.1% of women reported neither behavior. The proportion of neither behavior increased with age (Appendix A) except for women in China.

### 3.2. Prevalence of Heart Disease by Age, Gender, and Smoking/Drinking Behavior

Figure 2 shows the prevalence of heart disease with 95% CIs by age group and smoking and drinking behaviors by gender for both countries. Nonoverlapping 95% CIs between groups implies statistically significant differences (the prevalence and 95% CI can be found in Appendix A).

In the US, the overall prevalence of heart disease was higher in men (24.5%, 95% CI: 22.5–26.6%) than in women (20.6%, 95% CI: 19.3–22.1%), and the prevalence increased with age for men and women with greater gender gap. In China, the overall prevalence in women (22.9%, 95% CI: 21.7–24.1%) was higher than men (16.1%, 95% CI: 15.1–17.2%), the prevalence of heart disease decreased in the oldest age group for both men and women and gender gap decreased with age.

For both men and women in the US, the drink-only group had the lowest prevalence (men: 15.4%, 95% CI: 13.3–17.7%; women: 14.8%, 95% CI:12.7–17.2%), while the highest prevalence was found in the smoke-only group (men: 35.5%, 95% CI: 32.4–38.6%, women: 31.5%, 95% CI: 28.7–34.4%). In China, those who reported neither behavior had the lowest level (men: 15.4%, 95% CI: 11.8–19.8%; women: 21.7%, 95% CI: 20.4–23%); women who smoked only or engaged in both behaviors showed the highest prevalence of heart disease (~34.4%) but there was little fluctuation among men (~16%).

### 3.3. Prevalence Ratios of Heart Disease by Smoking/Drinking Behavior

Table 2 shows the weighted prevalence ratio (adjPR) of heart disease, 95% CIs, and *p*-values by age and smoking/drinking behavior, adjusting for education, diabetes, high blood pressure, sleep quality, and BMI categories. Crude PRs and adjusted PRs for all variables are presented in Appendix A.

In the US, smoke-only was associated with a higher prevalence of heart disease, the adjPR was 1.16 in men (95% CI: 1.01, 1.35, *p* = 0.046) and 1.34 in women (95% CI: 1.21, 1.49, *p* < 0.001) compared with those who did not smoke or drink. Drink-only was associated with lower prevalence of heart disease, the adjPR was 0.75 in men (95% CI: 0.63, 0.89, *p* < 0.001) and 0.97 in women (95% CI: 0.85, 1.11, *p* = 0.675).

In China, smoke-only was associated with heart disease among women (adjPR = 1.48, 95% CI: 1.29, 1.70, *p* < 0.001). The association in men was borderline significant (adjPR = 1.35, 95% CI: 0.94, 1.95, *p* = 0.103). In women who reported both smoking and drinking, the adjPRs were 1.54 (95% CI: 0.25, 1.89, *p* < 0.001) in women. The association between smoking/drinking behavior and heart disease in men had similar trend but results were not statistically significant.

## 4. Discussion

This cross-sectional study estimated the prevalence disparities of heart disease among older men and women (50 y+) with smoking/alcohol drinking behaviors in the US and China. In the US, the prevalence was higher in men (24.5%) than in women (20.6%) and increased with age for both genders. In contrast, women in China (22.9%) had higher prevalence than men (16.1%), and the 80+ age group had higher prevalence than the 70–79 age group. The association between smoking/drinking behavior and heart disease were different in the two countries. In the US, higher prevalence was found in the smoke-only group for both genders, while in China, the prevalence for women who smoked or engaged in both behaviors were ~1.5 times (95% CI: 1.3–1.8, *p* < 0.001) those who did not smoke or drink, but no statistical differences were found in men.

The pattern of the increasing prevalence of heart disease by age and gender differences found in the HRS sample was similar to the US National Center for Health Statistics [4]. The decreased prevalence of heart disease among the 80+ age group in CHARLS might partially be due to survival bias. The 80+ age group (born before 1936) in China had poor health status and lower life expectancy because they had experienced the most unstable society and survived from wars, social chaos, and famine [26]. Additionally, the sample sizes of age 80+ in the CHARLS were relatively small and only accounted for 5.8% of men (*n* = 340) and 7.1% of women (*n* = 418), in comparison to 9.3% (*n* = 1203) US older women and 12.9% (*n* = 1874) older men.

The opposite gender differences in China may be explained by historical background and the gender inequality. Previous literature showed that older women in China always had lower social status and were undereducated with lower income levels [7]. Similar statistics were also observed from our study that more than 90% of women in China had lower than high school educational level, and higher percentage of high blood pressure, as well as diabetes, were also observed among women when compared with men. Far fewer women than men in China reported smoking, which might be related to the health inequities and social backgrounds. In the traditional culture, smoking and drinking were discouraged for women, and women in the older generations needed to comply with the traditional gender role [27], although, with the rapid socioeconomic changes in the last few decades, women’s social status has improved, and such changes might also bring great pressure and poor mental health conditions for older women [28]. The large gender gaps observed in the smoke-only and both smoking/drinking behavior groups might be related to the reality that older women were more likely to be underinsured and lack the ability to access the social health insurance for self-health care [29]. The heavy impacts of smoking on heart disease among women in China found in our study indicate that to better control heart disease, future health care strategies should address such unhealthy behavior among women.

These findings support our hypothesis that smoking behavior is associated with a higher prevalence of heart disease, and the association differs by gender and country. Smoking was associated with heart disease prevalence with a stronger effect on women, especially for the women in China, which was consistent with a previous literature that women who smoked in China had higher mortality and worse prognosis after acute cardiovascular events [30]. The impact of smoking on heart disease was also reported by Tolstrup et al. [18]. They utilized pooled datasets and demonstrated that smoking control was essential for heart disease prevention, and the majority of heart disease was attributable to smoking for all age groups [18]. Therefore, the important impacts of smoking behavior on heart disease among older populations cannot be ignored, according to our findings, which elicits the recommendation that older populations in the US and China should be encouraged to reduce cigarette consumption to prevent and control heart disease and also improve their quality of life.

In addition, we found that drink-only is associated with lower prevalence of heart disease among US men. Abat et al. showed that alcohol consumption levels were associated with aging [31]. A few studies found that light to moderate alcohol consumption was safe and would be beneficial to the cardiovascular system [32,33]. However, several other researchers defended the harmful effects of alcohol intake, even at a low consumption level, stating that it outweighed their benefits [34,35]. Thus, the estimated effects of drinking might be misleading due to the unaccounted consumption amounts and frequency of drinking in this study. Therefore, it would be safe to recommend that which aligns with previous studies, to keep a light to moderate alcohol intake among the current drinkers and to continue no drinking for nondrinkers for health improvement and heart disease prevention [31,33,36].

The results of null association between smoking and heart disease among men in China were also observed in a study of cross-country comparison of cardiovascular disease attributable to tobacco exposure in China and the USA [37]. Smoking is a top risk factor for mortality that can cause various diseases such as cardiovascular disease and lung cancer [38,39]. The tobacco epidemic among men in China (71% smoking or smoking/drinking) could have a large effect on mortality, and therefore might produce the survival bias in the prevalence of heart disease.

This study was conducted based on the relatively large sample size from two national-level surveys in the US and China. The results would be comparable since this study utilized harmonized datasets with similar years in both countries, and the findings might be generalizable to other older populations. However, this study has several limitations. First, both surveys in the US and China were based on self-reported data, and thus might have information biases such as misclassification due to recall biases of specific chronic disease conditions. Although harmonized datasets were used, the specific drinking types and the educational level categories were not the same in both countries. Third, there were some potential but unmeasured confounding variables in this study, such as race, physical activity, mental health status, types and dosage of smoking and alcohol consumption, income level, and nutritional status. Physical activity was an important predictor of heart disease but was not included in this study. Future studies should consider these potential confounders to estimate the association disparities of heart disease among the old populations by country. Fourth, since the study design was cross-sectional, causal inferences are limited. Only prevalence of heart disease could be observed in the cross-sectional analyses while the incidence of heart disease and its association with smoking and alcohol drinking behaviors were unable to be determined in the current study. Cohort studies are needed to identify the impacts of smoking and alcohol drinking behaviors on heart disease incidence and mortality, especially for the comparisons between the older populations in the US and China.

## 5. Conclusions

Utilizing datasets from two national surveys in the US and China, our study illustrated the consistent patterns that indicate the prevalence of heart disease was associated with smoking among men and women in both countries. The findings from the study of the older population might form the baseline information for the development of earlier control and management of cigarette consumption, which would be an important strategy for health promotion and reducing the burden of heart disease in both the US and China. In China, while preserving the low level of smoking in women, more effective measures are needed to promote smoking cessation among younger men before development of fatal diseases.

## Figures and Tables

**Figure 1 ijerph-19-02188-f001:**
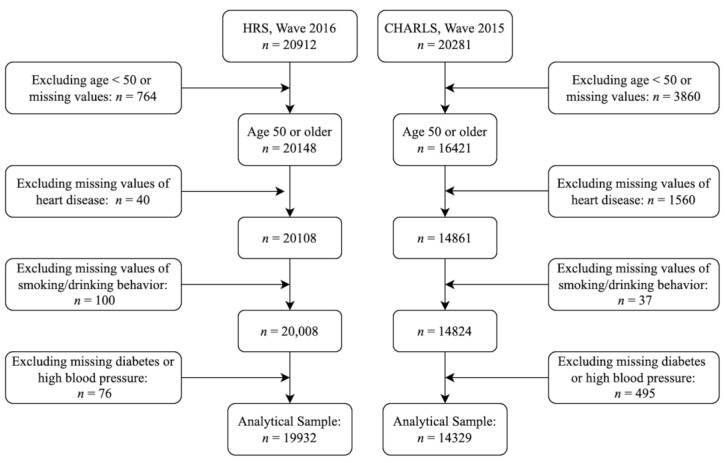
Analytical sample derived from 20,912 participants in the 2016 wave of HRS data and 20,281 participants in the 2015 wave of CHARLS data.

**Figure 2 ijerph-19-02188-f002:**
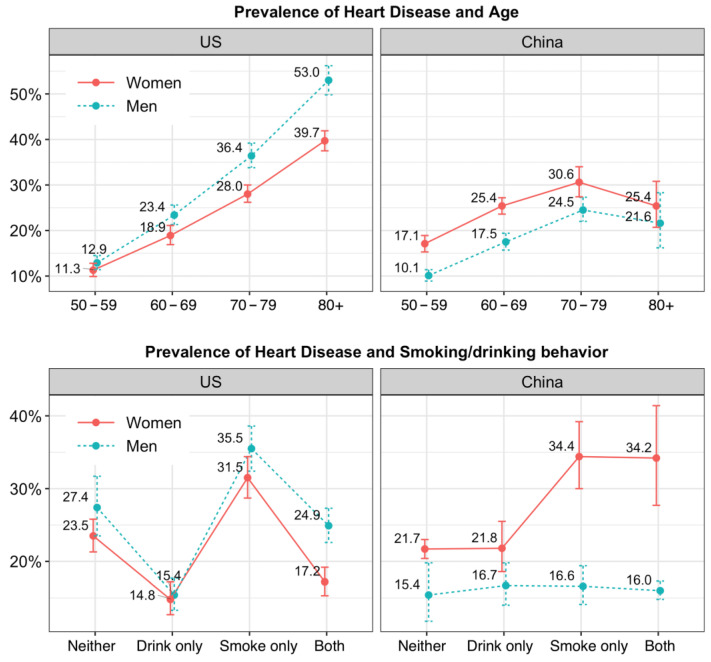
Weighted prevalence of heart disease by age and smoking/drinking behavior and corresponding 95% CI. Nonoverlapping 95% CIs between groups implies statistically significant differences.

**Table 1 ijerph-19-02188-t001:** Sample size (*n*) and weighted percentages (wt%) by gender and country.

	US (*n* = 19,932)		China (*n* = 14,329)	
	Men (46.6%)	Women (53.4%)	Men (48.7%)	Women (51.3%)
	*n* (wt%)	*n* (wt%)	*n* (wt%)	*n* (wt%)
Heart disease				
No	6190 (75.5)	8969 (79.4)	5833 (83.9)	5640 (77.1)
Yes	2205 (24.5)	2568 (20.6)	1166 (16.1)	1690 (22.9)
Age group (years)				
50–59	2980 (38.6)	3868 (34.8)	2695 (38.7)	2946 (40.2)
60–69	2492 (33.7)	3363 (33.5)	2671 (37.2)	2747 (35.7)
70–79	1720 (18.4)	2432 (18.8)	1293 (18.3)	1219 (17.0)
80+	1203 (9.3)	1874 (12.9)	340 (5.8)	418 (7.1)
Median and maximum	64 and 103	65 and 107	62 and 101	62 and 105
Smoking and drinking behavior				
Neither	1151 (12.8)	3204 (24.3)	449 (6.9)	5224 (71.1)
Only drinking	2019 (28.1)	2703 (26.5)	732 (10.2)	1380 (19.3)
Only smoking	1750 (18.4)	2283 (17.9)	1538 (21.8)	505 (6.7)
Both	3475 (40.7)	3347 (31.3)	4280 (61.1)	221 (2.9)
Education				
Less than high school	1460 (11.9)	1998 (12.5)	5836 (82.3)	6743 (91.7)
High school or equivalent	4207 (48.4)	6141 (51.6)	946 (14.5)	514 (7.0)
Associate degree or higher	2728 (39.7)	3398 (35.9)	217 (3.1)	73 (1.3)
Household income				
Tertile 1	2176 (21.4)	4437 (30.9)	1441 (19.5)	1596 (20.9)
Tertile 2	2897 (30.5)	3748 (30.5)	1499 (20.2)	1528 (19.8)
Tertile 3	3322 (48.1)	3352 (38.6)	1509 (22.4)	1550 (22.3)
Did not report	-	-	2550 (37.8)	2656 (37.0)
High blood pressure				
No	3236 (42.7)	4475 (45.3)	4551 (64.4)	4533 (61.9)
Yes	5159 (57.3)	7062 (54.7)	2448 (35.6)	2797 (38.1)
Diabetes				
No	6013 (75.2)	8479 (77.5)	6317 (90.2)	6430 (87.8)
Yes	2382 (24.8)	3058 (22.5)	682 (9.8)	900 (12.2)
Sleep quality				
Restless	1729 (21.1)	2901 (25.0)	1755 (24.4)	2998 (39.4)
Good	6614 (78.4)	8572 (74.6)	4839 (69.3)	3851 (52.7)
Did not report	52 (0.5)	64 (0.5)	405 (6.3)	481 (7.9)
BMI				
Normal	1779 (20.0)	3125 (29.3)	3517 (49.0)	3330 (45.5)
Overweight	3571 (43.6)	3563 (30.7)	1423 (20.8)	1856 (24.2)
Obese	2907 (35.1)	4368 (36.0)	191 (2.7)	397 (4.9)
Underweight	83 (0.8)	239 (2.1)	381 (5.1)	377 (5.1)
Did not report	55 (0.5)	242 (1.9)	1487 (22.5)	1370 (20.3)
Race/ethnicity				
White	4924 (73.9)	6628 (73.8)	-	-
Hispanic	1369 (10.0)	1785 (10.0)	-	-
Black	1650 (10.0)	2583 (11.2)	-	-
Other	452 (6.1)	541 (5.0)	-	-

**Table 2 ijerph-19-02188-t002:** Weighted prevalence ratio (adjPR) of heart disease, 95% confidence intervals (CI)s, and *p*-values by age and smoking/drinking behavior.

Country	Variables	Men		Women	
		adjPR * (95% CI)	*p*	adjPR * (95% CI)	*p*
US	Age (years)				
	50–59	Ref		Ref	
	60–69	1.55 (1.33, 1.81)	<0.001	1.51 (1.27, 1.79)	<0.001
	70–79	2.11 (1.84, 2.43)	<0.001	1.89 (1.63, 2.19)	<0.001
	80+	3.04 (2.64, 3.49)	<0.001	2.51 (2.14, 2.96)	<0.001
	Smoking/drinking behavior				
	Neither	Ref		Ref	
	Only drinking	0.75 (0.63, 0.89)	0.001	0.92 (0.78, 1.09)	0.309
	Only smoking	1.16 (1.01, 1.35)	0.046	1.34 (1.21, 1.49)	<0.001
	Both	1.02 (0.89, 1.16)	0.817	0.97 (0.85, 1.11)	0.675
China	Age (years)				
	50–59	Ref		Ref	
	60–69	1.68 (1.40, 2.01)	<0.001	1.36 (1.19, 1.55)	<0.001
	70–79	2.31 (1.95, 2.73)	<0.001	1.50 (1.30, 1.74)	<0.001
	80+	1.96 (1.40, 2.74)	<0.001	1.31 (1.04, 1.65)	0.024
	Smoking/drinking behavior				
	Neither	Ref		Ref	
	Only drinking	1.23 (0.87, 1.76)	0.246	1.01 (0.86, 1.18)	0.948
	Only smoking	1.35 (0.94, 1.95)	0.103	1.48 (1.29, 1.70)	<0.001
	Both	1.26 (0.91, 1.74)	0.162	1.54 (1.25, 1.89)	<0.001

* adjPR: weighed prevalence ratio of heart disease adjusting for BMI, household income, educational level, sleep quality, high blood pressure, diabetes, and BMI categories. For US sample, race/ethnicity was also included in the analysis.

## Data Availability

Data can be download from https://g2aging.org/ (accessed on 1 July 2021) by registered users.

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
