# Peer review of "Gender Disparities of Heart Disease and the Association with Smoking and Drinking Behavior among Middle-Aged and Older Adults, a Cross-Sectional Study of Data from the US Health and Retirement Study and the China Health and Retirement Longitudinal Study"

_ijerph, 2022, doi:10.3390/ijerph19042188_

Round 1
Reviewer 1 Report
Thank you for allowing me to review this manuscript. This manuscript titled "Gender disparities in heart disease and the association with smoking and drinking among middle-aged and older adults 3, a cross-sectional study of data from the US Health and Retirement Study 4 and the China Health and Retirement Study. 5 tudinal Study. ”This study aimed to estimate the prevalence of heart disease among 77 middle-aged and older adults (50 years or older) by gender, and to analyze their associations with smoking / drinking alcohol 78 in the United States and China.
It is an interesting and highly relevant article today, although it has several limitations that make it susceptible to publication in this journal. These limitations are detailed below:
- The material and methods section does not reflect different important ethical considerations. It would be necessary to indicate whether the data collection was authorized by an ethics committee. It is also not reflected if the participation of the patients was voluntary or if they were offered a document with the information and informed consent.
- The tables do not specify a footer for the acronyms used in them.
- The conclusions are clear and precise. However, due to the importance and topicality of the topic, it should include the impact of the results today. In addition, it would be interesting to include a future line.
- There are errors in the bibliographic references. Said references are not adapted to the regulations required by the journal. It is necessary for the publication of the manuscript to review said bibliographic references.
Reviewer 2 Report
The authors attempt to investigate gender disparities in prevalence of CV diseases and the association with smoking and drinking behaviour in US and Chinese cohorts. The main value of the study is the large number of patients included, for which the authors should be commended.
My main concern is that there is no attempt to quantify the amount of smoking or consumption of alcohol which could therefore potentially explain the observed sex differences. Therefore no definitive conclusions can be made as sex differences in the effects of smoking and alcohol consumption on CV disease. If the authors do not have this data, the best that they can get out it is to compare the population attributable risk for smoking & alcohol consumption between the men & women.
Reviewer 3 Report
In the reviewed article, the authors analyzed gender differences in the prevalence of heart disease depending on smoking and drinking behavior in middle-aged and older adults in the USA and China. This cross-sectional study used data from the US Health and Retirement Study and the China Health and Retirement Longitudinal Study. The authors indicated as the main finding of the study the association of the prevalence of heart disease with smoke-only among men and women in both countries. The subject of the manuscript fits well with the scope of special issue devoted to associations of lifestyle factors and chronic diseases. In my opinion, several issues need to be addressed in a revision of this paper. The major problems of manuscript are following:
- Too many confounding variables make it difficult to draw clear, explicit conclusions. Applying more restrictive inclusion/exclusion criteria (for example, the presence of comorbidities like diabetes, obesity, or hypertension) would result in more homogeneous samples and could facilitate the interpretation of results.
- The age structure in each smoking and drinking behavior group would be interesting.
- The discussion is very general and sketchy. The authors focused primarily on the reasons of differences in prevalence of heart disease between men and women in China and on enumeration the numerous limitations of the study. There are some other aspects that should be addressed, for example: Why are there no differences in prevalence of heart disease due to smoking / drinking behavior in Chinese men, or what could be the reason for lowering the prevalence of heart disease in the group of drinking and smoking patients (both) to the level of the control group (neither)?
- The conclusions are very general and add little to the health promotion.
Reviewer 4 Report
Thnak you for the opportunity to review this article.
Recommnedations>
Abstract
The conclusion is simplistic and does not highlight the main results identified by the study.
Introduction
I recommend extending the introduction and highlighting the novelty aspects of the study compared to previous studies.
Discussions
We recommend extending the discussions to highlight the correlations or differences identified by the main results of this study compared to previous research.
Conclussions
We recommend extending the conclusions in order to highlight the main ideas that emerge from the current research and possibly identify the practical perspectives.
Round 2
Reviewer 3 Report
The authors addressed all my concerns and revised the manuscript accordingly.
Reviewer 4 Report
The authors improved the manuscript according with the recommendations.